# Higher Body Fat but Similar Phase Angle Values in Patients with the Classical Form of Congenital Adrenal Hyperplasia in Comparison to a Control Group

**DOI:** 10.3390/nu14235184

**Published:** 2022-12-06

**Authors:** Núbia Maria de Oliveira, Raquel David Langer, Sofia Helena Valente Lemos-Marini, Daniel Minutti de Oliveira, Bruno Geloneze, Gil Guerra-Júnior, Ezequiel Moreira Gonçalves

**Affiliations:** 1Laboratory of Growth and Development (LabCreD), Center for Investigation in Pediatrics (CIPED), School of Medical Sciences (FCM), State University of Campinas (UNICAMP), Campinas, Sao Paulo 13083-887, Brazil; 2Laboratory of Investigation of Metabolism and Diabetes (LIMED), School of Medical Sciences (FCM), State University of Campinas (UNICAMP), Campinas, Sao Paulo 13083-878, Brazil

**Keywords:** bioimpedance, body composition, adrenal hyperplasia, BIVA, adults

## Abstract

This study aimed to compare phase angle (PhA) and bioelectrical impedance vector analysis (BIVA) values between adult patients with congenital adrenal hyperplasia caused by 21-hydroxylase deficiency (CAH21OHD) and a control group. A total of 22 patients (15 women, 22.9 ± 3.7 years) were compared with 17 controls (11 women, 27.0 ± 2.5 years). Body composition was determined by dual-energy X-ray absorptiometry. Bioelectrical impedance was used to calculate PhA, and BIVA was performed using specific software. Student’s *t*-test and analysis of covariance were used to compare groups. Hedges’ G and partial n^2^ were calculated for the effect estimates. Hotelling’s t^2^ test was used to compare the mean impedance vectors between the groups. The Mahalanobis test was used to determine the distance between confidence ellipses. Patients with CAH21OHD had a higher fat mass percentage than that of the control group (both sexes). There was no significant difference in PhA values between groups (CAH21OHD vs. control) in females (6.9° vs. 6.3°, *p* = 0.092) and males (8.2° vs. 8.1°, *p* = 0.849), after adjusting for covariates (age and height). BIVA analysis showed a significant difference in the mean impedance vectors between the female groups (T^2^ = 15.9, D = 1.58, *p* = 0.003) owing to the higher reactance/height (Δ = 8.5; *p* < 0.001) of the patients. The PhA did not significantly differ between the groups. Female patients had significantly higher reactance values. However, further studies are needed to determine the usefulness of bioimpedance parameters in evaluating the hydration status and cellular integrity of patients with CAH21OHD.

## 1. Introduction

Patients with congenital adrenal hyperplasia due to 21-hydroxylase deficiency (CAH21OHD) have impaired endogenous glucocorticoid production, leading to a reduction in cortisol feedback. Consequently, an increased release of adrenocorticotropic hormone from the pituitary gland and the excessive secretion of adrenal androgens and their precursors occurs [1]. The current therapy consists of glucocorticoid replacement and, in some cases, mineralocorticoids replacement [2]. Both under- and overtreatment result in unpleasant side effects and complications [1,2,3]. However, long-term exposure to glucocorticoids, particularly supraphysiological doses, is associated with high body fat, cardiovascular disease, insulin resistance, and osteopenia [3,4]. Thus, monitoring and evaluating the nutritional status of patients with CAH21OHD is important.

The bioelectrical impedance analysis (BIA) is a non-invasive, safe, and practical method that provides resistance (R) and reactance (Xc) parameters to evaluate body composition among different populations [5,6]. The Phase angle (PhA) is an index calculated from BIA parameters and has been used as a qualitative method to predict health among hospitalized patients and healthy individuals [6]. A recent systematic review discussed the relationship of PhA and obesity in adults. Significant differences in the PhA value of obese individuals in comparison to controls was note observed, except in those with severe obesity, who showed lower PhA values; moreover, a negative relationship between PhA and fat mass was observed [7]. Contrastingly, the PhA values among healthy men and women (38.7 ± 12.3 and 37.1 ± 12.1 years, respectively) were positively associated with running performance [8]. BIA can be complemented by bioelectrical impedance vector analysis (BIVA), which uses the values of R and Xc adjusted for height (h) plotted on a bivariate graph to produce an impedance vector with length and direction [9]. The individual location in the graph is interpreted according to (i) the position along the major axis of the ellipses, which indicates the body hydration state (i.e., long vector means less hydration; short vector means more hydration); and (ii) the position on the left or right side of the minor axis indicating, respectively, whether more or less body cell mass is contained in the soft tissues, and reflects the PhA [5,9]. The vectors of each individual can then be evaluated within the 50%, 75%, and 95% tolerance ellipses of a reference population, and the participant groups can also be evaluated using bivariate 95% confidence ellipses of the mean vectors [5,9]. BIVA is useful in monitoring changes in hydration status and body cell mass in patients undergoing hemodialysis treatment [10], at-risk patients with impaired nutritional and functional status [11], and patients with acute heart failure [12].

However, to date, no study has investigated the differences in BIA parameters between patients with CAH21OHD and healthy adults. In addition, only one study in children with CAH21OHD [13] verified the influence of factors related to body composition and glucocorticoid treatment on the PhA value. Therefore, this study aimed to compare PhA values and impedance vectors between adults with CAH21OHD and a control group (free from major diseases). CAH21OHD patients commonly have high body fat and an unfavorable metabolic profile due to complexities related to the balance between controlling the effects of the disease and adequate glucocorticoids dosage [3,4]. Despite that, the previous studies of our group did not find cardiometabolic risk profiles among patients with CAH21OHD [14,15]. Furthermore, there were no differences for PhA values according to different adiposity levels in children and adolescents with CAH21OHD [13]. Therefore, we hypothesized that CAH21OHD patients would have similar PhA values but different vector positions in BIVA when compared to the control group.

## 2. Materials and Methods

### 2.1. Design and Ethical Aspects

This was an observational case-control study to test the hypothesis of possible differences in BIA parameters of CAH21OHD patients in comparison to a control group. Considering that the disease has an incidence of approximately 1 to 15,000 births worldwide [1], and the difficulty in finding specially male patients in whom the effect of the disease is less perceptible [4], we sought to evaluate the maximum number of adult patients in the Clinical Hospital who met the inclusion criteria for this study. In addition, the control group was selected by convenience, which consisted of graduation and post-graduation students who volunteered to participate in the study. All of the evaluations were carried out in the same day and participants underwent laboratory assessments early in the morning. The assessments were carried out in the Metabolic Clinical Research Unit at the Laboratory of Investigation of Metabolism and Diabetes and in the Laboratory of Growth and Development, both situated at the School of Medical Sciences at UNICAMP, Brazil. The procedures were approved by the local ethics committee (number 768/2007) in accordance with the Declaration of Helsinki for studies involving humans. Informed consent was obtained from all the participants.

### 2.2. Study Participants

A total of 44 patients (age range: 18–31 years) from the Pediatric Endocrinology Outpatient Clinic at the State University of Campinas (UNICAMP) were invited to participate in the study. The inclusion criteria were (i) individuals with a confirmed diagnosis of the classic form of CAH21OHD based on clinical and hormonal criteria using molecular analysis [16,17,18,19]; and (ii) those who can visit the clinical hospital for routine follow-ups for at least 2 years. All patients were diagnosed at childhood as a result of virilization signs (sex ambiguity in females and precocious puberty in males), high serum levels of ACTH, 17OH-progesterone and androstenedione (and in salt-wasting form with high serum levels of renin and potassium and low levels of sodium) and were confirmed by CYP21A2 gene sequencing. Thus, 22 patients (15 women [22.9 ± 3.7 years] and seven men [23.8 ± 4.5 years]) were included in the data analysis. In addition, control group participants who volunteered to participate in the study were recruited through the researchers’ social media and phone. The inclusion criteria were (i) individuals with age similar to that of the CAH21OHD group, and (ii) those with an absence of chronic diseases, hormonal dysfunctions, and severe mental and psychiatric disorders. Thus, 17 healthy adults (11 women [27.0 ± 2.5 years] and six men [24.4 ± 2.3 years]) were included in the data analysis.

### 2.3. Anthropometry

Body weight (kg) was measured using a portable digital scale accurate to 0.1 kg, and height (cm) was measured using a vertical stadiometer accurate to 0.1 cm. Body mass index (BMI, kg/m^2^) was then calculated. The participants wore light clothing and were barefoot during all measurements.

### 2.4. Body Composition

Total body composition was calculated to determine the fat mass (FM, kg and %) and lean soft tissue (LST, kg) using a dual-energy X-ray absorptiometry device (iDXA, version 13.60, GE Healthcare Lunar, Madison, WI, USA), according to the manufacturer’s instructions. The lean soft tissue index (LSTI) was calculated by dividing the LST by height squared (LST/H^2^, kg/m^2^). The percentage of FM android (%FM_Android_) and gynoid (%FM_Gynoid_) was defined according to the equipment software. The android area is the region around the waist between the midpoint of the lumbar spine and top of the pelvis, and the gynoid area is the region between the head of the femur and mid-thigh.

### 2.5. Bioelectrical Impedance Analysis

BIA measurements were obtained using a single frequency tetrapolar device (50 kHz), and a Quantum II (RJL Systems, Detroit, MI, USA). The participants were instructed to maintain their normal routine of food and hydration and follow the recommendations described in the literature prior to the test [20]. The CAH21OHD patients were instructed to take their medication as usual. All the participants underwent a single assessment early in the morning in a fasting state. The volunteers were instructed to lie on a table in a supine position and were insulated from electrical conductors; arms were positioned apart without touching the body, and the legs were abducted at an angle of 45°. After 5 min of rest, the participants’ skin was cleaned with alcohol, and two electrodes were placed on the surface of each right hand and right foot. The device provided resistance (R) and reactance (Xc) values in ohms (Ω). The precision of the parameters provided by our BIA device was determined by the coefficient of variation (CV%) and technical error of measurement (TEM) based on the test-retest in our laboratory. The CV% for R and Xc was 0.35% and 0.33%, respectively, and the TEM was 3.54 Ω and 0.49 Ω, for R and Xc, respectively. The phase angle (PhA) was calculated using the following equation: PhA = arc-tangent(Xc/R)x(180/π) [21]. For PhA the %CV was 11.6 and the TEM was 0.07°. BIVA was performed correcting R and Xc measurements by the height of participants in meters (m), thus expressing both R/H and Xc/H in ohm/m (Ω/m) [9]. Tolerance ellipses of 50%, 75%, and 95% were generated by the BIVA software using a healthy population as a reference [22]. The length and position of the vectors provided information about the state of body hydration and cell mass contained in the soft tissues, respectively. A comparison between groups was performed using confidence ellipses in the same software [22]. The separate 95% confidence ellipses indicated a significant difference between the mean impedance vectors of the groups.

### 2.6. Glucocorticoid Therapy

Glucocorticoids were converted to hydrocortisone equivalent dosage (HDE) (20 mg hydrocortisone = 5 mg prednisone = 0.75 mg dexamethasone = 2 mg fludrocortisone) [23] using the dose prescribed during the year prior to the assessment. Additionally, daily HDE was indexed using the body surface area (m^2^) according to the DuBois and DuBois equation [24]. The data of glucocorticoid therapy were obtained from the medical records and confirmed with all of the patients. Androstenedione levels were measured using the Modular E170 automated chemiluminescent immunometric method (Roche Diagnostics, Indianapolis, IN, USA), with an intra-assay coefficient of variation < 3.5%. The routine glucocorticoid and mineralocorticoid therapies for each patient are detailed in Appendix A.

### 2.7. Statistical Analysis

SPSS version 25.0 (Statistical Package for the Social Sciences, Chicago, IL, USA) was used for the statistical data analysis. Variables that did not show a normal distribution were transformed using the logarithm of base 10 (Xc, %FMandroid, and R/H). The student’s t-test for independent samples was used to compare variables between the groups (patients and controls). An analysis of covariance was used to compare the PhA values between groups (patients and controls) controlled for age and height (female and male sexes). Hotelling’s t^2^ test was used to compare the mean impedance vectors between groups. The Mahalanobis test was used to calculate the distance between ellipses. For effect size (E.S) estimates, the G of Hedges and partial n^2^ were calculated. The E.S for variables with a non-normal distribution was calculated using the formula r = z/√n. An E.S of <0.19 was considered trivial; ≥0.20, small; ≥0.50, moderate; ≥80, large; and ≥1.30, very large [25]. For all analyses, statistical significance was set at *p* < 0.05.

## 3. Results

Women with CAH21OHD were younger (*p* = 0.004; E.S = 1.23), shorter (*p* = 0.029; E.S = 0.89), and had higher %FM (*p* = 0.027; E.S = 0.91) and %FM_android_ (*p* = 0.045; E.S = 0.92) than women in the control group. Men with CAH21OHD were shorter (*p* = 0.012; E.S = 1.55) and had higher %FM (*p* = 0.048; E.S = 1.15) and higher %FM_gynoid_ (*p* = 0.048; E.S = 1.15) than men in the control group (Table 1).

A comparison of bioimpedance parameters (Table 2) revealed that women with CAH21OHD had higher values of Xc, Xc/H, and PhA (*p* = 0.002, E.S = 0.58; *p* = 0.001, E.S = 1.47; and *p* = 0.013, E.S = 1.03, respectively) than those in the control group. In male participants, no significant differences were found in the BIA parameters between the groups (*p* > 0.05).

Figure 1 shows the PhA values between groups separated by sex and adjusted for age and height (females) and height (males). There was no significant difference in the PhA values between groups (CAH21OHD patients vs. control) in both sexes (females: 6.9° [95% CI: 6.5–7.5] vs. 6.3° [95% CI: 5.7–6.9], *p* = 0.092; E.S = 0.12 and males: 8.2° [95% CI: 7.3–9.1] vs. 8.1° [95% CI: 7.1–9.1], *p* = 0.849; E.S = 0.004).

Figure 2 shows the individual vector analysis (tolerance ellipses) and comparison of the mean vectors of the groups (confidence ellipses) separated by sex. In females (Figure 2A), greater dispersion was observed in CAH21OHD patients than in the control group. Women with CAH21OHD were positioned in the upper pole, five were positioned outside the 75% limits (within 95% tolerance), and four were positioned above the 95% limits. Men with CAH21OHD were located at the upper pole. One male patient was positioned outside the 75% limits (within the limits of 95%), and four patients were positioned outside the 95% tolerance limits in the upper region of the graphic. In addition, five men with CAH21OHD had vectors that shifted to the left of the major axis (Figure 2B). Among the females (Figure 2C), the separated ellipses showed a significant difference in the mean impedance vectors between the groups (D = 1.58, *p* = 0.003). The ellipses of women with CAH21OHD were above the ellipse of the control group owing to a higher Xc/H ratio; contrastingly, R/H and PhA were slightly higher than for those in the control group. Among males (Figure 2D), the overlapping ellipses show no significant difference in the mean vectors between the CAH21OHD patients and the control group (D = 1.08, *p* = 0.224).

## 4. Discussion

This study’s main and novel findings are that no differences were found in the PhA values between adults with CAH21OHD and the control group, even after adjusting for confounding variables. However, the amount and distribution of FM differed between groups of both sexes. Additionally, significant differences were observed in the impedance vectors between women with CAH21OHD and the control group.

The higher adiposity observed in patients with CAH21OHD in the present study has been reported in previous studies [26,27]. Moreover, we observed that patients with CAH21OHD had a higher body fat distribution (%FM_android_ and %FM_gynoid_ in women and men, respectively) than that of the control group. FM in the android region is associated with greater cardiovascular risk [28]; in addition to the elevated fat levels, patients with CAH21OHD commonly have unfavorable lipid, glycemic, and cardiovascular profiles [27,29]. Nevertheless, previous studies of our group did not find cardiometabolic risk profiles among patients with CAH21OHD [14,15]. Glucocorticoid therapy contributes to an increase in body FM in this population [3,4]; however, overweight and obesity are observed in individuals receiving both physiological and supraphysiological doses [4]. Other factors, such as impaired adrenomedullary function (decreased secretion of adrenaline, a hormone that contributes to lipolysis and reduced insulin secretion) and changes in the leptin axis (an important regulator of energy balance), play a key role in the development of obesity in these patients [4,30,31].

Furthermore, the higher body fat observed in the CAH21OHD group may have been reflected in the BIA parameters owing to higher R-values compared to that of the control group (although not statistically significant, trivial E.S = 0.21 and moderate E.S = 0.51, in women and men, respectively). The R-value reflects body hydration, given that body FM is a poorly hydrated tissue and will present greater resistance to the passage of the electric current (resulting in a higher R-value) [6,32]. Conversely, the amount of lean soft mass present in the body greatly influences the R-value because the muscle mass has large amounts of water and electrolytes, thus offering less resistance to the passage of electric current [32]. However, in our study, we found no difference in the LST between patients with CAH21OHD and the control group, even after adjusting for height (LSTI) (women: *p* = 0.638; E.S = 0.18; men: *p* = 0.508; E.S = 0.35) and the correlation observed between R/H values and LSTI were very close in both groups (CAH21OHD: r = −0.89 *p* ≤ 0.001, control group: r = −0.93 *p* ≤ 0.001, data not shown). In addition, FM did not show a negative influence on the PhA value in patients with CAH21OHD compared to that of the control group. One previous study carried out in children with CAH21OHD, from both sexes and according to different levels of adiposity (tertiles of %FM), did not find a significant difference in the PhA values [13]. However, it was found that the body composition and the glucocorticoid dosage were determinants of the PhA value. For example, among girls, the variables of lean soft tissue, the glucocorticoid dosage, and height (R^2^ = 0.68, *p* < 0.001) were the determinant factors for the PhA value, while among boys, age, glucocorticoid dosage, and FM (R^2^ = 0.82, *p* < 0.001) were the determinant factors for the PhA value [13].

Patients with CAH21OHD in the present study had PhA values similar to those reported in previous studies that used a large sample of healthy adults (20–29 years) with BMIs between 18.5–26 kg/m^2^ [33,34]. This indicates that CAH21OHD did not negatively affect the PhA value in our study-enrolled participants. Thus, it appears that patients with CAH21OHD showed a state of cellular integrity and hydration similar to those in the aforementioned studies [33,34]. This finding may reflect adequate hormonal control in these patients, due to routine follow-ups, supervisions and glucocorticoid dosage close to the physiological needs. In addition, other studies with part of the sample from the present study have already reported that these patients do not present a negative picture in health-related aspects, such as cardiometabolic, lipid, and bone health profiles [14,15,35]. In healthy cells, part of the electric current that penetrates the capacitive element of cell membranes is delayed, which creates a phase shift quantified geometrically as the PhA [36]. Although the biological meaning of PhA is not completely understood, it is a strong indicator of cell membrane integrity and function mainly related to the ICW/ECW ratio [6]. In this sense, higher PhA values are strong and positively related with higher ICW/ECW, whereas lower PhA values reflects higher ECW, which may indicate cell damage, inflammation, loss of body cell mass and worse clinical condition [11,37].

In the BIVA analysis, female tolerance ellipses showed all patients with CAH21OHD in the upper pole, with 33% falling outside the 75% area (inside the 95% ellipse) and 26% falling outside the 95% area. Five patients were positioned beyond the 75% area, with vectors shifted to the right of the major axis, indicating less hydration and lower body cell mass [6,38]. According to Moore et al. 1963 [39], body cell mass reflects the metabolically active component of the human body, and the main fractions are the component cells present in the muscles and viscera [39,40]. These patients showed low lean mass compared with those positioned on the left side of the ellipse (Appendix A). Furthermore, the confidence ellipse showed a significant difference in the mean vectors of the CAH21OHD group compared with that of the control group. The plot in Figure 2C shows that CAH21OHD patients seem to have a tendency to higher vector length, which may indicate less TBW [38], although what determined the difference between groups was the higher Xc/H value (48.2 ± 6.8 vs. 39.7 ± 3.3; *p* = 0.001) with a high E.S of 1.47 in the CAH21OHD group.

Xc reflects the volume of the cell membrane capacitance and intracellular content; in fact, it was demonstrated that higher Xc values explained higher values of ICW [41]. It is expected that intact cell membranes will act as capacitors by storing the electric current and releasing it [36]. In this sense, Xc and PhA were closely and positively related, and both were indicators of the cell membrane integrity [11,42]. In addition, Xc and PhA were negatively related to muscle damage [42]. In addition, there are other factors that can influence the cell capacitance (Xc value), such as cell size, composition, thickness, the distance between cell membranes, and the distribution of cellular fluids (intra/extracellular) [21,43]. Moreover, the amount of lean soft tissue can influence the Xc value [11,38]; however, in the present study, we found no difference in the amount of LSTI between the groups. Furthermore, no correlation (r = 0.09, *p* = 0.680, data not shown) between the amount of body fat in patients and the Xc value was found. In this sense, owing to the lack of studies using bioimpedance parameters in this population, it was difficult to determine which characteristics could have contributed to the high Xc value in patients with CAH21OHD. However, it can be speculated that the glucocorticoid and mineralocorticoid (responsible for changing the balance of water and sodium, carbohydrate metabolism, lipids, and immunity) therapies [44] may influence the Xc value. Additionally, the mean Xc values reported by the present study participants were not similar to the mean reported by a previous study that used a large sample of healthy women (range, 20–29 years) [45]; hence, since this is the first study of BIA parameters in patients with CAH21OHD, it was difficult to determine which values would be within the expected range for this sample.

In males, BIVA analysis showed that all patients with CAH21OHD were positioned in the upper pole, whereas 57% fell outside the 95% area, and 71% of the patients with CAH21OHD had vectors shifted to the left of the major axis, indicating a higher body cell mass and high PhA value [5,6,38]. Two patients with CAH21OHD showed longer vectors lying outside the 95% limits on the right side of the minor axis, indicating less hydration and cell mass [38] (Appendix A). Despite the not statistically significant differences in confidence ellipses, the men with CAH21OHD followed the same pattern of women, and also showed higher vector length (Figure 2D), suggesting that these patients may have less TBW in relation to the control group. The BIVA analysis considers both the PhA value (related to the minor axis) and the vector length (major axis) [6,38]; individuals with the same PhA value may show differences in body composition that can be identified by the BIVA graph through the length of the vector [11]. In addition, the graphical view from BIVA allows us to check which parameters (R or Xc) interfere most in the PhA value. This tool might be useful for assessing and monitoring patients with CAH21OHD, providing information related to body cell mass, cellular integrity and hydration status, as has already been reported in previous studies among people with different health conditions [5,11,12]. Nonetheless, it should be pointed that in a previous study of CAH21OHD children and adolescents [13], PhA was not sensitive to the differences in body fat levels; in fact, a systematic review concluded that differences in PhA are noticeable only when the excess of fat is very marked [7]. Moreover, the vector length in classic BIVA (used in our study) only provides information about the hydration status. In addition, to test if BIVA could be useful to detect changes caused by weight loss or gain, further follow up studies would be necessary.

A limitation of this study is the small number of participants, particularly males, in whom the effect of the disease is less noticeable [1,4], as this might had influenced the statistical power of comparisons. Considering our variables of interest (BIA parameters) in females, the statistical power was between 29 and 90%, and among males, the statistical power was very low (not exceeding 30%). Nonetheless, it is important considering the incidence of the disease, and that studies with CAH21OHD patients commonly have a low number of participants. Another point is the not very well-matched controls, especially in the female sex where there was a significant difference in age; despite that, we used statistical adjustments in the analysis. In addition, the participants were all graduation and post-graduation students and were instructed to follow normal routines including food, hydration and physical exercise in the days approaching the tests. The strength of this study is that patients with CAH21OHD were monitored for a long period (21 ± 4.02 years) in the same hospital and by the same team of physicians, thus presenting a low bias in relation to the treatment. It is worth commenting on the low variability in the measures (R, Xc and PhA) of our instrument demonstrated by the %TEM of 0.73, 0.77, and 7.4 for R, Xc and PhA, respectively. This suggests that the differences observed between groups in females for Xc (mean % difference = 15.6) and Xc/H (21.5%) were not influenced by a possible lack of precision of our instrument. In addition, this is the first study providing information on raw BIA parameters in a sample of patients with CAH21OHD. In summary, this study provides novel and important information that needs to be highlighted: (i) the PhA, when adjusted for covariates (age and height), suggested a similar intra/extracellular fluid distribution and cellular health between patients with CAH21OHD and individuals free from major diseases; and (ii) according to the BIVA graph, the high value of Xc/H in women with CAH21OHD suggests that the disease and treatment do not adversely affect the patients’ cellular integrity in this study. However, further studies are needed to determine the effectiveness of BIA parameters in evaluating and monitoring hydration status and cellular integrity in patients with CAH21OHD.

## Figures and Tables

**Figure 1 nutrients-14-05184-f001:**
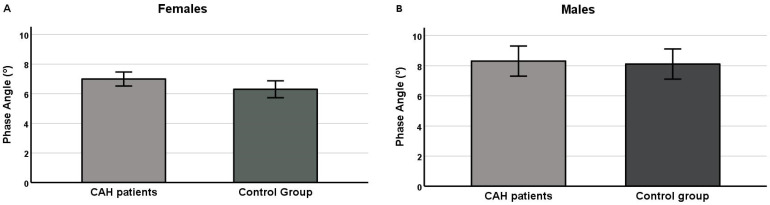
Estimated means of the phase angle value of groups and according to sex. Females (**A**): adjusted for age (24.61 years) and height (157.5 cm); F = 3.109, *p* = 0.092; E.S = 0.12. Males (**B**): adjusted for height (166.30 cm); F = 0.038 *p* = 0.849; E.S = 0.004.

**Figure 2 nutrients-14-05184-f002:**
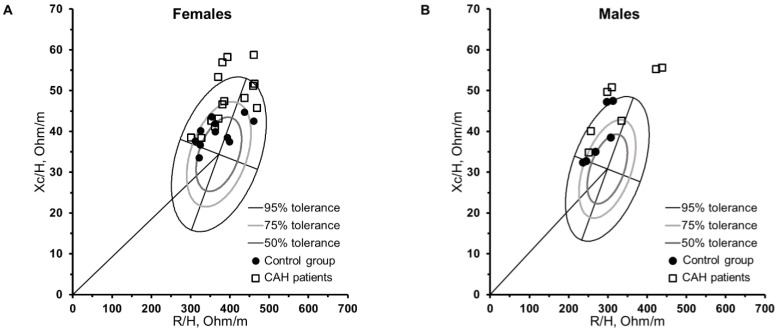
Vector analysis for the comparison of study participants (CAH21OHD group and control group) in female (**A**) and male (**B**) tolerance ellipses and confidence ellipses in females (**C**) and males (**D**).

**Table 1 nutrients-14-05184-t001:** Descriptive characteristics and body composition separated by sex and group of the study participants.

	Females				Males			
	Control (n = 11)	Patients (n = 15)			Control (n = 6)	Patients (n = 7)		
	Mean ± SD	Mean ± SD	*p*-Value	E.S	Mean ± SD	Mean ± SD	*p*-Value	E.S
DGT (years)		22.3 ± 3.6				19.1 ± 4.3		
HDE (mg/m^2^/day)		13.2 ± 4.8				12.5 ± 2.9		
Age (years)	27.0 ± 2.5	22.9 ± 3.7	**0.004**	1.23	24.4 ± 2.3	23.8 ± 4.5	0.771	0.15
Weight (kg)	58.7 ± 8.9	60.8 ± 13.6	0.654	0.17	71.0 ± 10.8	65.4 ± 12.9	0.415	0.44
Height (cm)	161.3 ± 7.1	154.8 ± 7.0	**0.029**	0.89	173.7 ± 7.6	160.0 ± 8.7	**0.012**	1.55
BMI (kg/m^2^)	22.6 ± 3.0	25.3 ± 4.8	0.106	0.65	23.4 ± 1.7	25.5 ± 4.6	0.319	0.54
%FM	31.6 ± 5.6	37.2 ± 6.3	**0.027**	0.91	20.3 ± 4.9	29.4 ± 8.9	**0.048**	1.15
LST (kg)	37.81 ± 5.5	35.5 ± 4.8	0.270	0.43	53.9 ± 7.9	43.7 ± 8.9	0.052	1.12
LSTI (kg/m^2^)	14.5 ± 1.3	14.9 ± 2.1	0.610	0.20	17.8 ± 1.3	17.0 ± 2.7	0.518	0.34
%FM_Android_	1.7(1.0–3.1) ^a^	2.8(1.0–4.4) ^a^	**0.025**	0.92	1.7(0.7–2.1) ^a^	2.8(0.8–3.9) ^a^	0.088	0.77
%FM_Gynoid_	6.4 ± 0.9	7.9 ± 1.2	0.233	0.37	3.3 ± 1.0	5.2 ± 1.8	**0.048**	1.15

DGT, duration of glucocorticoid therapy; HDE, hydrocortisone dose equivalent; BMI, body mass index; %FM, fat mass percentage; LST, lean soft tissue; LSTI, lean soft tissue index; ES, effect size. ^a^ Median (min-max). Values in boldface are statistically significant (*p* < 0.05).

**Table 2 nutrients-14-05184-t002:** Bioelectrical impedance parameters among patients with CAH21OHD and the control group separated by sex.

BIA Parameters	Females				Males			
	Control (n = 11)	Patients (n = 15)	*p*-Value	E.S	Control (n = 6)	Patients (n = 7)	*p*-Value	E.S
	Mean ± SD	Mean ± SD			Mean ± SD	Mean ± SD		
R (Ω)	592.7 ± 68.2	610.4 ± 91.6	0.095	0.21	480.2 ± 40.8	503.2 ± 95.4	0.346	0.51
R/H (Ω/m)	362.5 (311.7–461.0) ^a^	380.4 (300.3–468.7) ^a^	0.226	0.48	282.6 (235.7–312.8) ^a^	309.8 (251.1–437.6) ^a^	0.146	0.39
Xc (Ω)	64.0 (57.0–68.0) ^a^	74.0 (57.0–95.0) ^a^	**0.002**	0.58	64.5 (59.0–78.0) ^a^	75.0 (56.0–88.0) ^a^	0.206	0.70
Xc/H (Ω/m)	39.7 ± 3.3	48.2 ± 6.8	**0.001**	1.47	38.9 ± 6.9	47.0 ± 8.0	0.078	1.00
PhA (°)	6.2 ± 0.7	7.1 ± 0.8	**0.013**	1.03	8.0 ± 0.7	8.3 ± 1.0	0.590	0.29

BIA, bioelectrical impedance parameters; R, resistance; Xc, reactance; H, height; PhA, phase angle. ^a^ Median (min-max). Values in boldface are statistically significant (*p* < 0.05).

## Data Availability

Not applicable.

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
