# Peer review of "Higher Body Fat but Similar Phase Angle Values in Patients with the Classical Form of Congenital Adrenal Hyperplasia in Comparison to a Control Group"

_nutrients, 2022, doi:10.3390/nu14235184_

Round 1

Reviewer 1 Report

The authors compare raw BI measurements and BIVA characteristics among patients with congenital adrenal hyperplasia (CAH) relative to healthy control adults.  Although resistance values (absolute and normalized for height) were not significant within each gender group, reactance and phase angle were near significant in women with CAH. The authors should consider the following comments.

Additional details of the experimental design would be informative. For example, clearly state the experimental hypothesis and provide calculations to determine sample size to test the hypothesis.

The association of phase angle with obesity is an increasingly important issue. Please refer to Di Vincenzo O et al. Clin Nutr 2021; 40(9): 5238-5248.

Additional description of the BIA device is needed. Describe how you determined the validity of the BIA device (e.g., technical accuracy and precision). Include the technical error of measurement of resistance, capacitance and phase angle.

The BIA data reported in Table 2 clearly indicate significant differences (greater values) in Xc (74 vs 64 ohm)m, Xc/H (39.7 vs 48.2 ohm/m) and PhA 6.2 vs 7.1 degress) in the female CAH patients compared to female controls. Different values are cited throughout the manuscript.

It is important to discuss the physiological interpretation of Xc and PhA. Refer to reference 37; Lukaski HC et al. Nutrients 2019;11(4):809; Francisco R et al. Int J Environ Res Public Health 2020;24;17(3):759.

Comment of the reported differences in BIA measurements as compared with the technical error measurement of these impedance variables.

It is noteworthy that similar differences in BIA measurements of CAH patients by gender group are shown in Table. This observation suggests larger sample sizes in females (e.g., higher statistical power) compared to males may explain the lack of significance in the males. 

Additional discussion of the components in the BIVA plots is needed. For example, vector length appears to be longer in the female and male CAH patients. This findings indicates a lower TBW in the CAH patients. Discuss the relationship between vector length and TBH (see reference 37).

Author Response

Reviewer 1

Comments and Suggestions for Authors

The authors compare raw BI measurements and BIVA characteristics among patients with congenital adrenal hyperplasia (CAH) relative to healthy control adults.  Although resistance values (absolute and normalized for height) were not significant within each gender group, reactance and phase angle were near significant in women with CAH. The authors should consider the following comments.

 Author response: The authors are grateful to the reviewer for the thoughtful comments. We feel the suggestions have improved the paper and we have made our changes in the original manuscript (in blue). Responses to each of the specific comments are given below.

Additional details of the experimental design would be informative. For example, clearly state the experimental hypothesis and provide calculations to determine sample size to test the hypothesis.

Author response: The experimental hypothesis description was amended as requested to include additional information of the study and now it reads: “CAH21OHD patients commonly have high body fat and unfavorable metabolic profile due to complexities related to the balance between controlling the effects of the disease and adequate glucocorticoids dosage 3,4. Despite that, the previous studies of our group did not find cardiometabolic risk profiles among patients with CAH21OHD14,15, also, there was no differences for PhA value according to different adiposity levels in children and adolescents with CAH21OHD13. Therefore, we hypothesized that CAH21OHD patients would have similar PhA values but different vector positions in BIVA when compared to the control group.” In addition, we added more details about the experimental design and justified the lack of a sample size calculation in our methods section and now it reads: “This was an observational case-control study to test the hypothesis of possible differences in BIA parameters of CAH21OHD patients in comparison to a control group. Considering that the disease has an incidence of approximately 1 to 15.000 births worldwide1, and the difficulty of find especially male patients, in whom the effect of the disease is less perceptible4, we sought to evaluate the maximum number of adult patients in the Clinical Hospital who met the inclusion criteria for this study. In addition, the control group was selected by convenience, which consisted of graduation and post-graduation students who volunteered to participate in the study. All the evaluations were carried out in the same day and participants underwent laboratory assessments early in the morning.’’

The association of phase angle with obesity is an increasingly important issue. Please refer to Di Vincenzo O et al. Clin Nutr 2021; 40(9): 5238-5248.

Author response: The paper mentioned above was cited in the introduction section where it now reads: “A recent systematic review discussed the relation of PhA and obesity in adults. It was not observed significant differences in the PhA value of obese individuals in comparison to controls, except in those with severe obesity, which showed lower PhA values, moreover, it was observed a negative relation of PhA and fat mass7.”

Additional description of the BIA device is needed. Describe how you determined the validity of the BIA device (e.g., technical accuracy and precision). Include the technical error of measurement of resistance, capacitance and phase angle.

Author response: Changes were made in the methods section. This section now reads: “The precision of the parameters provided by our BIA device was determined by the coefficient of variation (CV%) and technical error of measurement (TEM), based on the test-retest in our laboratory. The CV% for R and Xc was 0.35% and 0.33%, respectively, and the TEM was 3.54 Ω and 0.49 Ω, for R and Xc, respectively.” We also added the values for PhA: ‘’For PhA the %CV was 11.6 and the TEM was 0.07°.’’

The BIA data reported in Table 2 clearly indicate significant differences (greater values) in Xc (74 vs 64 ohm)m, Xc/H (39.7 vs 48.2 ohm/m) and PhA 6.2 vs 7.1 degress) in the female CAH patients compared to female controls. Different values are cited throughout the manuscript.

Author response: Changes have now been made throughout the manuscript.

It is important to discuss the physiological interpretation of Xc and PhA. Refer to reference 37; Lukaski HC et al. Nutrients 2019;11(4):809; Francisco R et al. Int J Environ Res Public Health 2020;24;17(3):759.   

Author response: The discussion about the PhA meaning was amended in the discussion section and now it reads: ‘‘In healthy cells, part of the electric current that penetrates the capacitive element of cell membranes is delayed, which creates a phase shift quantified geometrically as the PhA37. Although the biological meaning of PhA is not completely understood, it is a strong indicator of cell membrane integrity and function mainly related to the ICW/ECW ratio6. In this sense, higher PhA values are strong and positively related with higher ICW/ECW, whereas lower PhA values reflects higher ECW, which may indicate cell damage, inflammation, loss of body cell mass and worse clinical condition12,38.’’ In addition, the discussion about the Xc was complemented, and now it reads: ‘‘Xc reflects the volume of the cell membrane capacitance and intracellular content, in fact, it was demonstrated that higher Xc values explained higher values of ICW41. It is expected that intact cell membranes will act as capacitors by storing the electric current and releasing it 36. In this sense, Xc and PhA were closely and positively related, and both were indicators of the cell membrane integrity11,42, also, Xc and PhA were negatively related with muscle damage42.’’

Comment of the reported differences in BIA measurements as compared with the technical error measurement of these impedance variables.

Author response: Changes were made in the discussion section where it now reads: ‘‘It is worth commenting the low variability in the measures (R,Xc and PhA) of  our instrument demonstrated by the %TEM of 0.73, 0.77, and 7.4 for R, Xc and PhA, respectively. Which seems that did not influence the significant differences between groups observed in females for Xc (mean % difference=15.6), Xc/H (21.5%) and PhA before the adjustment (14.51%), strengthening the ability to demonstrate the differences found in these parameters between groups.’’

It is noteworthy that similar differences in BIA measurements of CAH patients by gender group are shown in Table. This observation suggests larger sample sizes in females (e.g., higher statistical power) compared to males may explain the lack of significance in the males.  

Author response: The limitation of the sample size in males was considered and discussed in the last paragraph of the discussion section: ‘‘A limitation of this study is the small number of participants, particularly males, in whom the effect of the disease is less noticeable 1,4, this might had influenced the statistical power of comparisons. Considering our variables of interest (BIA parameters) in females, the statistical power was between 29 and 90%, and among males, the statistical power was very low (not exceeding 30%).’’

Additional discussion of the components in the BIVA plots is needed. For example, vector length appears to be longer in the female and male CAH patients. This findings indicates a lower TBW in the CAH patients. Discuss the relationship between vector length and TBH (see reference 37).

Author response: Additional information regarding  the vector length was included in the discussion section and now it reads: “The plot in figure 2C shows that CAH21OHD patients seem to have a tendence to higher vector length, which may indicate less TBW38, although what determined the difference between groups was the higher Xc/H value (48.2 ± 6.8 vs 39.7 ± 3.3; p=0.001) with a high E.S of 1.47 in CAH21OHD group.” The information about males was also included, and now it reads: “Besides, not statistically significant differences in confidence ellipses, the men with CAH21OHD followed the pattern of women, and also showed a tendence to higher vector length (figure 2D), suggesting that these patients may have less TBW in relation to the control group.”

Reviewer 2 Report

This paper addresses gaps in the BIA and BIVA literature pertaining to congenital adrenal hyperplasia. A couple of things surprised me about this analysis that might give ideas how to finalize the data presented.  

First surprise, the analysis shows only continuous measures, even though there are potential U-shaped effects in this kind of data.

Second surprise, the analysis finds higher Rx in the group with more body fat, even though several papers in the literature report the opposite. Lower Rx/h associated with overweight and obesity and higher serum sodium.

To put the results of this paper in context with other literature, it would help if the authors could show  more scatterplots of the data, like Rx/h on the y-axis and body fat measures on the x-axis and discuss the acknowledge and/or explain the opposite results across papers.

The paper might go further to show/suggest how to use the data in a way that clinicians might use the data to monitor these patients for weight gain. Is the BIA or BIVA data sensitive to small differences in health within the case group?

The discussion might mention complexities potentially related to 1) not very well matched controls, 2) who were the controls and what condition were they in?---Were they dehydrated by overnight fluid restriction or hot weather? 3) U-shapes related to slightly too much or too little sodium, water, too much too little medication?

Since the results are relative, cases relative to controls, it is important to understand if there is anything to know about the controls.

The text in the methods section might start with the overall study design. Move up that paragraph before the study participants paragraph.

It would help to have the criteria for defining CAH21OHD listed out instead of only the references [14-17]. In case the papers describe multiple options, it would be clearest to know exactly which criteria were used in this analysis.

Please add some sentences in the methods section about the protocol for data collection. What was the timing of the measurement? On a single day, at any particular time? Were participants given any instructions (like no food or drink, exercise, sleep) about how to prepare on the day or week before the data collection?

How were the data on glucocorticoid therapy captured? Medication use only collected for the patients? By self-report? Or by chart abstraction?

The most interesting point for clinical use of the BIA or BIVA might have something to do with the greater dispersion, more BIA or BIVA variability for cases. How to go deeper with that finding?

Also, to highlight where BIA or BIVA might not be useful..  e.g. if there is no difference between lower and higher body fat by phase angle.

Author Response

Comments and Suggestions for Authors

This paper addresses gaps in the BIA and BIVA literature pertaining to congenital adrenal hyperplasia. A couple of things surprised me about this analysis that might give ideas how to finalize the data presented.  

First surprise, the analysis shows only continuous measures, even though there are potential U-shaped effects in this kind of data.

Second surprise, the analysis finds higher Rx in the group with more body fat, even though several papers in the literature report the opposite. Lower Rx/h associated with overweight and obesity and higher serum sodium.

To put the results of this paper in context with other literature, it would help if the authors could show  more scatterplots of the data, like Rx/h on the y-axis and body fat measures on the x-axis and discuss the acknowledge and/or explain the opposite results across papers.

Author response: The authors are grateful to the reviewer for the thoughtful comments. We feel the suggestions have improved the paper and we have made our changes in the original manuscript (in blue). Responses to each of the specific comments are given below.

Despite the higher R/h median in CAH21OHD patients, this difference was not statistically significant, which could be due the low number of participants affecting the power of the comparisons (as discussed in our limitation section). Another possibility is because the R/h parameter is more related to body hydration, which is closely related to lean mass, and as showed in table 1, there was no significant difference in lean soft tissue index (LSTI) between groups. In addition, person’s correlation showed a significant correlation of R/h with LSTI in CAH21OHD (r= -0.89 p=<0.001) and control group (r= -0.93 p=<0.001). This information was added in the discussion section and now it reads: ‘‘and the correlation observed between R/H values and LSTI were very closely in both groups (CAH21OHD: r= -0.89 p=<0.001, control group: r= -0.93 p=<0.001, data not shown)’’. When we plotted the R/h on the y-axis and %FM on x-axis, there was a positive and significant correlation just in the control group (r=0.65, p=0.004), in CAH21OHD we did not find any tendency and Pearson’s r was 0.20, p=0.351. It is important to highlight that although these patients have higher body fat in relation to the control group the majority are not obese.

The paper might go further to show/suggest how to use the data in a way that clinicians might use the data to monitor these patients for weight gain. Is the BIA or BIVA data sensitive to small differences in health within the case group?

Author response: Changes were made in the discussion section discussing about PhA and BIVA as tools for monitoring these patients and now it reads: “This tool might be useful for assessing and monitoring patients with CAH21OHD giving information related to body cell mass, cellular integrity and hydration status as has already been reported in previous studies among people with different health conditions 5,11,12. Nonetheless, it should be pointed that with a previous study of CAH21OHD children and adolescents13, PhA was not sensitive to the differences in body fat levels, in fact, a systematic review concluded that differences in PhA are noticeable just when the excess of fat is very marked7. Moreover, the vector length in classic BIVA (used in our study) just give information about the hydration status. In addition, to test if BIVA could be useful to detect changes caused by weight loss or gain, further follow up studies would be necessary.’’

The discussion might mention complexities potentially related to 1) not very well matched controls, 2) who were the controls and what condition were they in?---Were they dehydrated by overnight fluid restriction or hot weather? 3) U-shapes related to slightly too much or too little sodium, water, too much too little medication?

Author response: The control group was selected by convenience. We found a significant difference in the age among female participants thus, we used statistical adjustments in the analysis. In addition, all the participants were graduation and post-graduation students and were instructed to follow normal routine (food, hydration, and physical exercises) while the collection was occurring. We discussed more about the control group in the discussion section, and now it reads: ‘‘…Another point is the not very well-matched controls, especially in the female sex where there was a significant difference in age, despite that, we used statistical adjustments in the analysis. In addition, the participants were all graduation and post-graduation students and were instructed to follow normal routine including food, hydration and physical exercise in the days near the tests.’’ We also emphasized that they were instructed to follow the recommendations described in literature for BIA assessments in the methods section, and now it reads: ‘’The participants were instructed to maintain normal routine of food and hydration and follow the recommendations described in the literature prior the test 19. The CAH21OHD patients were instructed to take their medication as usual. All participants underwent a single assessment early in the morning in fasting state.’’

Since the results are relative, cases relative to controls, it is important to understand if there is anything to know about the controls.

Author response: The information about the control group was added in the text.

The text in the methods section might start with the overall study design. Move up that paragraph before the study participant’s paragraph.

Author response: Changes were made in the methods section and the paragraph was moved, as requested.

It would help to have the criteria for defining CAH21OHD listed out instead of only the references [14-17]. In case the papers describe multiple options, it would be clearest to know exactly which criteria were used in this analysis.

Author response: We included the information to clarify the definition of CAH21OHD. This now reads as:All patients were diagnosed at childhood due to virilization signs (sex ambiguity in females and precocious puberty in males), high serum levels of ACTH, 17OH-progesterone and androstenedione (and in salt-wasting form with high serum levels of renin and potassium and low levels of sodium) and confirmed by the CYP21A2 gene sequencing”.

Please add some sentences in the methods section about the protocol for data collection. What was the timing of the measurement? On a single day, at any particular time? Were participants given any instructions (like no food or drink, exercise, sleep) about how to prepare on the day or week before the data collection?

Author response: These details have now been included in methods section: “The participants were instructed to maintain normal routine of food and hydration and follow the recommendations described in the literature prior the test 19. The CAH21OHD patients were instructed to take their medication as usual. All participants underwent a single assessment early in the morning in fasting state.”

How were the data on glucocorticoid therapy captured? Medication use only collected for the patients? By self-report? Or by chart abstraction?

Author response: The data of glucocorticoid therapy were obtained from the medical records and confirmed with all the patients. We added this information in the methods section.

The most interesting point for clinical use of the BIA or BIVA might have something to do with the greater dispersion, more BIA or BIVA variability for cases. How to go deeper with that finding?

Author response: The large dispersion of BIA and BIVA are factors that may interfere, but based on the calculated reproducibility and precision values, it seems that it was not a factor that interfered in the results. This information was included in the discussion section, where it now reads: ‘‘It is worth commenting the low variability in the measures (R,Xc and PhA) of  our instrument demonstrated by the %TEM of 0.73, 0.77, and 7.4 for R, Xc and PhA, respectively. Which seems that did not influence the significant differences between groups observed in females for Xc (mean % difference=15.6), Xc/H (21.5%) and PhA before the adjustment (14.51%), strengthening the ability to demonstrate the differences found in these parameters between groups.’’

Also, to highlight where BIA or BIVA might not be useful..  e.g. if there is no difference between lower and higher body fat by phase angle.

Author response: The information highlighting where BIA and BIVA might be useful was included in the discussion section. This now reads: “This tool might be useful for assessing and monitoring patients with CAH21OHD giving information related to body cell mass, cellular integrity and hydration status, as has already been reported in previous studies among people with different health conditions 5,12,13. Nonetheless, it should be pointed that with a previous study of CAH21OHD children and adolescents13, PhA was not sensitive to the differences in body fat levels, in fact, a systematic review concluded that differences in PhA are noticeable just when the excess of fat is very marked7. Moreover, the vector length in classic BIVA (used in our study) just give information about the hydration status. In addition, to test if BIVA could be useful to detect changes caused by weight loss or gain, further follow up studies would be necessary.’’

Round 2

Reviewer 1 Report

The authors adequately addressed my comments and noteworthy is their detailed description of the information in the RXc plot. The identification of hypohydration and perhaps mild dehydration as a factor explaining the differences in body fatness is important.